

# Bioprospecting of soil-borne microorganisms and chemical dereplication of their anti-microbial constituents with the aid of UPLC-QTOF-MS and molecular networking approach

Adivhaho Khwathisi,  Ntakadzeni Edwin Madala,  Afsatou Ndama Traore and Amidou Samie

Biochemistry and Microbiology, University of Venda for Science and Technology, Thohoyandou, South Africa

## ABSTRACT

Due to the emergence of drug-resistant microorganisms, the search for broad-spectrum antimicrobial compounds has become extremely crucial. Natural sources like plants and soils have been explored for diverse metabolites with antimicrobial properties. This study aimed to identify microorganisms from agricultural soils exhibiting antimicrobial effects against known human pathogens, and to highlight the chemical space of the responsible compounds through the computational metabolomics-based bioprospecting approach. Herein, bacteria were extracted from soil samples and their antimicrobial potential was measured via the agar well diffusion method. Methanolic extracts from the active bacteria were analyzed using the liquid chromatography quadrupole time-of-flight mass spectrometry (LC-QTOF-MS) technique, and the subsequent data was further analyzed through molecular networking approach which aided in identification of potential anti-microbial compounds. Furthermore, 16S rRNA gene sequencing enabled identification of the active bacterial isolates, where isolate 1 and 2 were identified as strains of *Bacillus pumilus*, whilst isolate 3 was found to be *Bacillus subtilis*. Interestingly, isolate 3 (*Bacillus subtilis*) displayed wide-ranging antimicrobial activity against the tested human pathogens. Molecular networking revealed the presence of Diketopiperazine compounds such as cyclo (D-Pro-D-Leu), cyclo (L-Tyr-L-Pro), cyclo (L-Pro-D-Phe), and cyclo (L-Pro-L-Val), alongside Surfactin C, Surfactin B, Pumilacidin E, and Isarrin D in the *Bacillus* strains as the main anti-microbial compounds. The application of the molecular networking approach represents an innovation in the field of bio-guided bioprospection of microorganisms and has proved to be an effective and feasible towards unearthing potent antimicrobial compounds. Additionally, the (computational metabolomics-based) approach accelerates the discovery of bioactive compounds and isolation of strains which offer a promising avenue for discovering new clinical antimicrobials. Finally, soil microbial flora could serve an alternative source of anti-microbial compounds which can assist in the fight against emergence of multi-drug resistance bacterial pathogens.

Corresponding authors
Ntakadzeni Edwin Madala,
ntaka.madala@univen.ac.za
Amidou Samie,
samie.amidou@univen.ac.za

## INTRODUCTION

Globally, drug resistance and re-emerging pathogens have been a growing concern in the agriculture, food, and medical industry, necessitating efforts to discover new antibiotics and enhance treatment options (*Shrivastava, Shrivastava & Ramasamy, 2018*). The unprecedented rate of drug resistance has been attributed to indiscriminate use of antibiotics, inadequacies on the part of the manufactures and acquired resistance through microbial evolution (*Podolsky, 2018*). Though some bacteria are naturally resistant to antibiotics, other bacteria gain resistance through different mechanisms such as mutation in the genes, acquisition of new genetic elements encoding protective enzymes and protein (*Bengtsson-Palme, Kristiansson & Larsson, 2018*). As such, while the number of microorganisms gaining resistance grows, so should the need to discover new antibiotics to address the evolving challenges posed by infectious diseases. Conversely, problems with rediscovery of known strains and compounds led to the decline in the discovery efforts during the second half of the 20th century, hence making it less attractive for pharmaceutical industry (*Atanasov et al., 2021*).

However, soil contains a high diversity of microorganisms (*Nannipieri et al., 2003*), some of which are essential for supporting life (*Mhlongo et al., 2018*). Nevertheless, our current understanding of soil-borne microbial populations remains limited, representing only a fraction of their actual diversity and composition, and their chemical space remains an untapped territory often described as "dark matter" in metabolomics (*Schultz et al., 2023*). Consequently, a holistic understanding of the metabolites and various compounds produced by these microorganisms, including those classified as "known unknowns", "unknown knowns", and "unknown unknowns" remains a mystery (*Hoskisson & Seipke, 2020*). Thus, a notable research gap exists, which warrants concerted efforts to elucidate the composition of microbial populations in soil and their associated chemical characteristics. Unlocking the full potential of microorganisms necessitates innovative and efficient approaches for the systematic exploration of their metabolic capabilities. Such efforts not only enhance our understanding of soil microbiomes but also offer potential in developing novel antimicrobial agents, contributing to the fight against antimicrobial resistance.

Recently, researchers have used several combinations of interdisciplinary approaches for bioprospecting, such as bioassays (*Cushnie et al., 2020*), genome mining (*Machado et al., 2015*) and metagenomics (*Thakur et al., 2020*). Although these approaches provided a good foundation for microbial bioprospecting, their limitation is their inability to provide insights into how these genes are regulated and how they interact within the microbial community (*Pierce & Dutton, 2022*). However, the emergence of omics science including metabolomics, has presented significant opportunities to expedite the screening and discovery of novel compounds and microbial strains (*Karwehl & Stadler, 2016*; *Mohana et al., 2018*). These advanced techniques enable comprehensive metabolite profiling, high-throughput screening, and the integration of genomics data, facilitating the discovery of previously unknown antimicrobial compounds (*Mohana et al., 2018*; *Maghembe et al., 2020*; *Aborode et al., 2022*).

Moreover, the resulting data from the advanced instruments is also complex and impossible to analyze with the traditional statistical methods. As such, multivariate statistical data models are used to decipher the inherited biological properties of these bioactive compounds by allowing an easy and computerized means of compound identification (*Tugizimana, Piater & Dubery, 2013*). Additionally, an alternative approach to decipher the chemical data acquired through these analytical instruments, particularly mass spectrometry, is molecular networking (*Aron et al., 2020*) which enables the grouping of metabolites based on their similar fragmentation patterns owing to common structural moieties (*Ramabulana et al., 2021*). Molecular networking allows for easier visualization of complex data and identification of unknown compounds through their association with known chemical entities within a given sample (*Clements et al., 2021*). This technique has been positively applied to isolate anti-microbial compounds from plant growth promoting bacterial strains (*Nephali et al., 2022*).

Herein, we present a methodological systematic approach for an effective bioprospection using a bio-guided molecular networking approach. Methanolic extracts of bacteria strains isolated from agricultural soils were analyzed using a combination of ultra-high performance liquid chromatography with quadrupole time-of-flight mass spectrometry (UPLC-Q-TOF-MS) and molecular networking approach to identify possible anti-microbial compounds thereof. Moreover, identification of the strains showing positive anti-microbial activities was achieved through 16S rRNA gene sequencing. The findings of the current study highlight the use of (computational metabolomics tool) molecular networking approach as feasible approach for the identification of bacterial strains capable of producing anti-microbial compounds. These compounds may thus be valuable for managing drug-resistant pathogens in the future. Moreover, molecular network approach allowed for a quick and easy-to-visualize approach for complex liquid chromatography with tandem mass spectrometry (LC-MS-MS) data.

## MATERIALS & METHODS

### Sample collection

Soil samples were gathered from 40 distinct locations within an agricultural setting in Tshakhuma village under Makhado Municipality (23°04′58.9″S 30°18′05.4″E), each at varying distances. The samples were obtained from each site at a depth of 0–20 cm below the surface without specific criteria. Upon collection, the samples were placed in sterile plastic bags and stored in darkness at 4 °C, then transported to the Microbiology research laboratory at the University of Venda for analysis. Processing of the samples was completed within 24 hours of their receipt. This text includes portions previously published as part of a thesis (*Khwathisi, 2021*).

### Selection of antibiotic producing strains

In the present study, the soil sprinkle technique was used to isolate microorganisms capable of producing antibiotics, following a protocol previously described by *Amin et al. (2012)*. To achieve this, 1 mg of soil particles were sprinkled on the surface of non-selective nutrient agar plates that had been previously inoculated with the test organisms,

*Staphylococcus aureus* ATCC (25923). The plates were then incubated aerobically at 37 °C for 24 hours. Antibiotic activity was assessed by observing the presence of zones of inhibition, surrounding a colony. Colonies displaying clear zones of inhibition around them were picked and streaked on separate nutrient agar plates to get pure cultures. Each isolated soil microorganism was assigned with specific codes and these isolates were used as the source of antibiotic producing microbes. Subsequently, all strains were preserved at −20 °C in 20% glycerol Muller Hinton Broth (Merck, Kenilworth, NJ, USA) for further confirmation and characterization by PCR and with periodic subculturing.

## Primary screening of antimicrobial compounds

Microorganisms isolated from different soil samples were screened for their antimicrobial activities. The test bacteria that were used for screening were *Staphylococcus aureus* ATCC (25923), *Pseudomonas aeriginosa* ATCC (9027), *Klebsiella pneumonia* ATCC (BAA-1705) and *Escherichia coli* ATCC (35218). Antimicrobial activities were assessed using Muller Hinton agar. Each plate was spread with the test organism and a small amount of the bacterial isolate was spotted on the test organism. Isolates displaying clear zones of inhibition against the pathogens were isolated onto plates and those displaying little or no inhibitory properties were re-tested and discarded if scarce antibiotic biosynthesis persisted. Purified microorganisms obtained were further tested for growth-inhibitory properties by using the agar well diffusion assay (*Woappi, Gabani & Singh, 2013*), against all the test microorganisms. Isolates that showed inhibitory zones against one or more pathogens were further purified and maintained at −20 °C in 20% glycerol Muller Hinton Broth (Merck, Kenilworth, NJ, USA).

## Molecular identification of the bacterial isolates

Pure bacterial isolates were cultured in Mueller Hinton Broth for 24 hours. Genomic DNA was extracted from bacterial pellets using the QIAamp DNA Mini Kit (Qiagen, Hilden, Germany) according to the manufacturer's instructions. Polymerase chain reaction (PCR) amplification of the 16S rRNA gene was performed using the primers 27F (5′-AGAGTT TGATCMTGGCTCAG-3′) and 1492R (5′-CGGTTACCTTGTTACGACTT-3′) as described by *Chen, Zhou & Gu (2015)*. Each PCR reaction of 25 μl in total included 12.5 μl DreamTaq DNA polymerase mix (Thermo Fisher Scientific, Waltham, MA, USA), 6.5 μl distilled water (dH$_2$O), the final concentration 0.4 μM of each primer and 4 μl DNA template. The cycling conditions were as follow: 5 min at 95 °C, 25 cycles at 94 °C for 40 s, at 55 °C for 30 s and 1 min at 72 °C, then followed by a final elongation step for 7 min at 72 °C. The PCR amplicons were sequenced by Inqaba-Biotech (Pretoria, South Africa), using the afore-mentioned PCR primers.

## Metabolite extraction

Pure bacterial isolates cultured in Mueller Hinton Broth for 24 hours were used for bacterial cell harvesting. A volume of 1 mL from the stock solutions was inoculated into Erlenmeyer flasks each containing 200 mL of nutrient broth. Erlenmeyer flasks were used to culture the bacterial isolates in triplicates and after 24 hours bacterial cells were harvested through centrifugation for 15 min at 5,000 rpm at 4 °C, resulting in supernatants and pellets

 

which were transferred into new conical tubes (50 mL). The damp weight of pellets was measured prior to extraction of the intracellular and extracellular metabolites by adding 80% LC-grade methanol (Romil SpS, Cambridge, UK). Consequently, the methanolic extract of the bacterial isolate were subjected to overnight agitation on a digital tube rotator at constant velocity maintained at 70 rpm to aid extraction. Centrifugation of the extracts was carried out for 15 min at 5,100 rpm at 4 °C. Therefore, the supernatant and methanolic extracts from the pellets were filtered using a 0.22 μm nylon filters into LC-MS glass vials as described elsewhere (*Ramabulana et al., 2021*).

## Liquid chromatography mass spectrometry analysis

Bacterial methanolic extracts analysis was performed using a liquid chromatography-quadrupole time-of-flight tandem mass spectrometer (LC-MS-9030 q-TOF; Shimadzu Corporation, Kyoto, Japan) fitted with a Shim-pack Velox C18 column (100 mm × 2.1 mm with particle size of 2.7 μm) as previously described by *Tlhapi et al. (2024)*. Specifically, the column oven temperature was maintained at 50 °C. The injection volume was 5 μL, and the samples were analytically separated over a 30 min binary gradient. The flow rate was kept constant at 0.3 mL/min in the current study using a binary solvent mixture of water with 0.1% formic acid (Eluent A) and methanol with 0.1% formic acid (Eluent B). The solvent gradient was set from 3 to 30 minutes to facilitate the separation of the compounds within the samples. Briefly, Eluent B was kept at 5% from 0 to 3 minutes, gradually increased from 5 to 40% between 3 and 8 minutes, and finally increased to 40–95% between 8 and 23 minutes. Eluent B was then kept isocratic at 95% between 23 and 25 minutes. The gradient was returned to original conditions of 5% at 25–27 minutes, and re-equilibration at 5% occurred at 27–30 minutes. The liquid chromatographic eluents were then subjected to a Quadruple Time-of-Flight high-definition mass spectrometer for analysis in positive electrospray ionization (ESI) mode. The q-TOF-MS conditions were as follows: 400 °C heat block temperature, 250 °C desolvation line (DL) temperature, 42 °C flight tube temperature, and 3 L/min nebulization and dry gas flow. Data was acquired through the data-dependent acquisition (DDA) mode, generating MS1 and MS2 data in tandem for ions falling within a mass-to-charge ratio (m/z) range of 100–2,000 and an intensity threshold above 1,000. Additionally, MS2 experiments employed argon gas for collision and maintained a collision energy of 30 eV with a spread of 5, utilizing sodium iodide (NaI) as a calibration solution to monitor high mass precision.

## Metabolite annotation and molecular networking

Molecular networks were generated on the Global Natural Product Social Molecular Networking (GNPS) platform (http://gnps.ucsd.edu). Prior to uploading to the online workflow, LC-qTOF-MS/MS raw data was converted to an open-source format (.mzML). Once spectral data was uploaded, it was processed by filtering out all MS2 fragment ions that were within ± 17 Da of the precursor m/z. Only the top four fragment ions in the ± 50 Da window were selected in order to window filter MS/MS spectra. An MS-CLUSTER technique was used to cluster the data. The MS2 fragment ion and a mass tolerance for precursor ion were both set at 0.05 Da. Only when more than six

matching peaks were present and 0.7 minimum cosine score was exceeded did edges (interconnections between metabolites/similarity) develop. In the network, only nodes that were each other's top 10 similar nodes were kept, which had a TopK value of 10. The maximum number of nodes that may be joined into a single molecular family was limited to 100. The spectral extracted from the mzML file was used to search through the GNPS in-house databases including the Mass Spectral database (MassBank), Human metabolome Database (HMBD) and the National Institute of Standards and Technology (NIST). Cytoscape software was used to visualize the generated molecular networks from GNPS (*Wang et al., 2016*; *Aron et al., 2020*). Molecular networks were extended by computing in silico annotation through network annotation propagation (NAP), which utilizes molecular network topology, predicts spectra fragmentation in silico, gets, and rates candidate structures. Substructure annotation was conducted using MS2 latent Dirichlet allocation (MS2LDA) within the GNPS interface. The outputs from molecular networking (MN), network annotation propagation (NAP) and MS2 latent Dieichlet allocation (MS2LDA) was combined by MolNetEnhancer which performed chemical classification. All matched and some mismatched nodes were confirmed or annotated based on empirical formulae derived from accurate mass and fragmentation patterns collected from MS2 studies. These annotated metabolites were also compared to available literature and searched against common dereplication databases for natural products such as KNApSAck (http://www.knapsackfamily.com/knapsack_core/top.php) and Dictionary of Natural Products (http://dnp.chemnetbase.com/faces/chemical/ChemicalSearch.xhtml). Metabolite annotation was completed at Metabolomics Standards Initiative (MSI) level 2. (*Sumner et al., 2007*).

## RESULTS

### Isolation of bacteria and antimicrobial activity testing

Out of 40 samples, only three isolates showed convincing activity against the ATCC test organisms. The colony morphology of these isolates varied, showing circular, irregular, and filamentous structures. The morphology and the size of the colonies varied from 1–10 mm in diameter with a relatively smooth surface at the beginning of the growth and the color ranged from white to creamy in pigmentation. Additionally, preliminary screening of soil isolates was carried out at 37 °C against the four ATCC strains. Maximum inhibition was achieved against *Staphylococcus aureus* ATCC (25923) by Isolate 3. Minimum inhibitory activity was noted against *Escherichia coli* ATCC (35218) by Isolate 1 as shown in Fig. 1. Testing for growth-inhibitory properties was done using the agar well diffusion assay under standard culturing conditions at 37 °C. The diameter of the growth inhibition zones around each well was measured and was taken to be related to the susceptibility of the antimicrobial substances produced by isolate to the test organism, and the diffusion rate of the antimicrobial substance through the agar medium.

### 16S rDNA aided identification of bacterial isolates

For genome extraction, three bacterial isolates that demonstrated efficacy against pathogenic organisms were selected. Using a nanodrop spectrophotometer, the presence

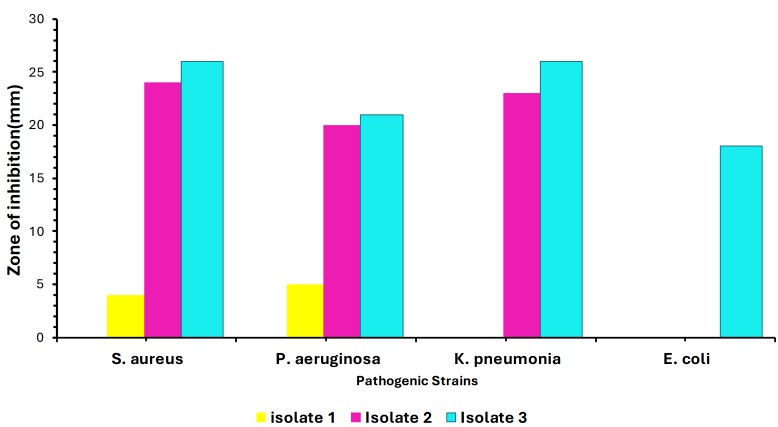

**Figure 1** **Infographic display of bacterial isolates extracts analyzed by agar diffusion technique against four pathogenic bacteria showing antimicrobial activity after 48 hours of incubation.** Colors represent the isolates tested, with NC being the negative control.

of DNA yield was evaluated (Table S2). The DNA was then utilized as templates for PCR amplification that followed. In the present study, the 16S rRNA gene sequences of three active isolates were determined. The Basic Local Alignment Search Tool (BLAST) was employed, and the findings revealed that all three bacterial isolates belonged to the genus *Bacillus*. The identities of the three isolates were determined by comparing them to the available data in the Genbank (NCBI) and with high-scored rRNA gene sequences in the BLAST searches. BLAST search similarities scores range between 99–100%. Phylogenetic analysis of the amplified 16S rRNA gene revealed that the two active isolates, Isolate 1 and Isolate 2 share sequence homology *Bacillus pumilus* (Fig. 2), on the other hand, Isolate 3 had convincing sequence homology with *Bacillus subtilis* (Fig. 3).

## UPLC-QTOF-MS analyses

A holistic untargeted metabolome analysis of antimicrobial producing bacteria was conducted. Bacterial cultures were grown on Mueller Hinton broth for 24 hours. To determine whether these bioactive metabolites in the crude extracts feature some unique chemical diversity, the methanol extracts of the strains were analyzed through ultra-high liquid chromatography quadruple time of flight mass spectrometry (UPLC-QTOF-MS) operating in positive ESI (+) mode. The spectrum data was acquired using the data-dependent acquisition (DDA) method, which simultaneously generates MS2 fragmentation data for every precursor ion over a predetermined threshold (*Ramabulana et al., 2021*). Figure S1 illustrates the base peak chromatogram of the samples indicating some similarities and differences in the metabolite profiles of the isolated *Bacillus* strains. Furthermore, these isolates showed to produce more extracellular than intracellular metabolites. Figure S1 shows that extracellular metabolites are more hydrophilic, whereas intracellular metabolites have a scarcity of hydrophilic compounds. To further investigate the chemical diversity of the bacterial isolates, the spectral data were analyzed through a publicly available database *via* the classical molecular networking workflow on the GNPS platform, enabling a broad

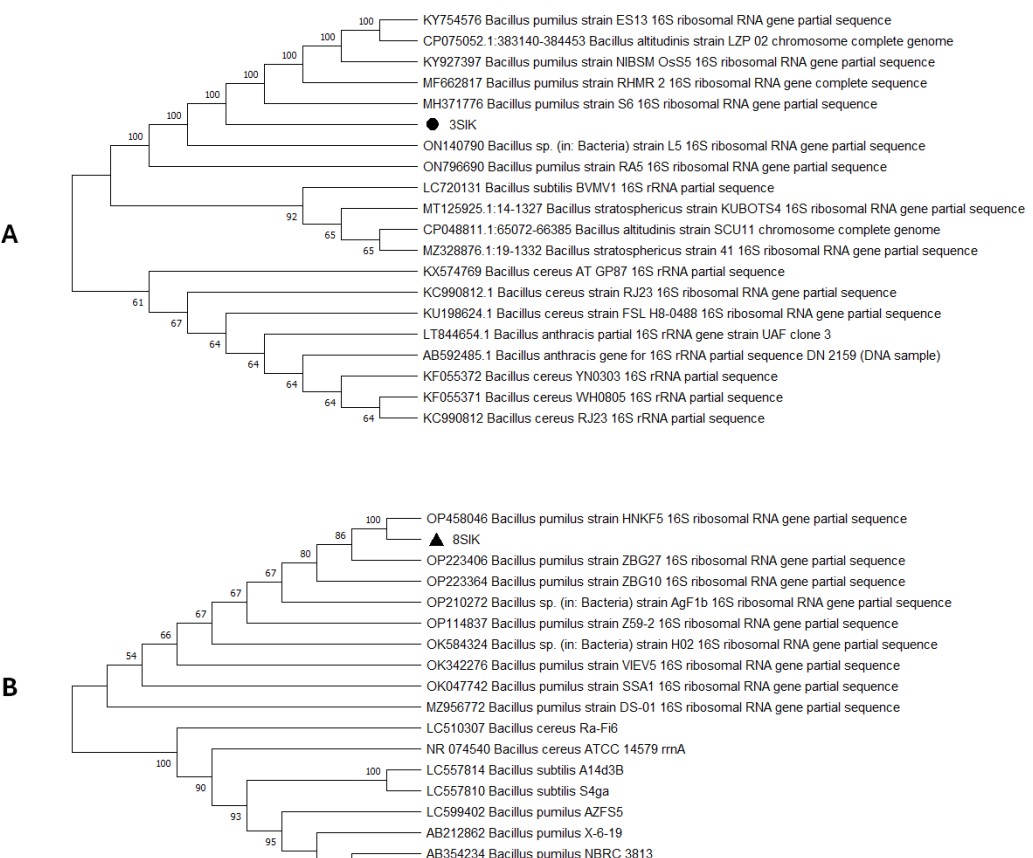

**Figure 2** **Phylogenetic tree based on 16s RNA gene fragment.** The neighbor-joining technique was used to generate the tree. (A) In this study, the black triangles and dot represents soil isolates. Isolate 1 (3SIK) with the 19 DNA sequences randomly chosen from the GenBank. (B) Isolate 2 (8SIK) with 18 DNA sequences chosen at random from GenBank and assigned accession numbers. The analyses of evolution were conducted using MEGA X (10.0.5) (*Tamura, Stecher & Kumar, 2021*).

overview of the chemical information deduced from the MS/MS data and the generated molecular networks were visualized in Cytoscape (Fig. S5) (*Doncheva et al., 2018*). The computed classical molecular network resulted in 1325 nodes, with 831 grouped into 22 independent molecular families (with at least two nodes connected by an edge) based on GNPS spectral matching. Insights into the chemical identities of the bacterial species were revealed by the computed molecular network (MN), which revealed seven of the nodes that were putatively annotated using an automatic library spectral matching against the public spectral libraries within the GNPS platform. It was also evident that not every bioactive metabolites (lipopeptides) were clustering in the same cluster (Fig. 4), simply indicating that there are some significant differences in the MS/MS fragmentation of these metabolites. Additionally, these isolates produce a diverse range of unidentified compounds which could be vital for the antimicrobial activities observed against test organisms and beyond (Fig. 1).

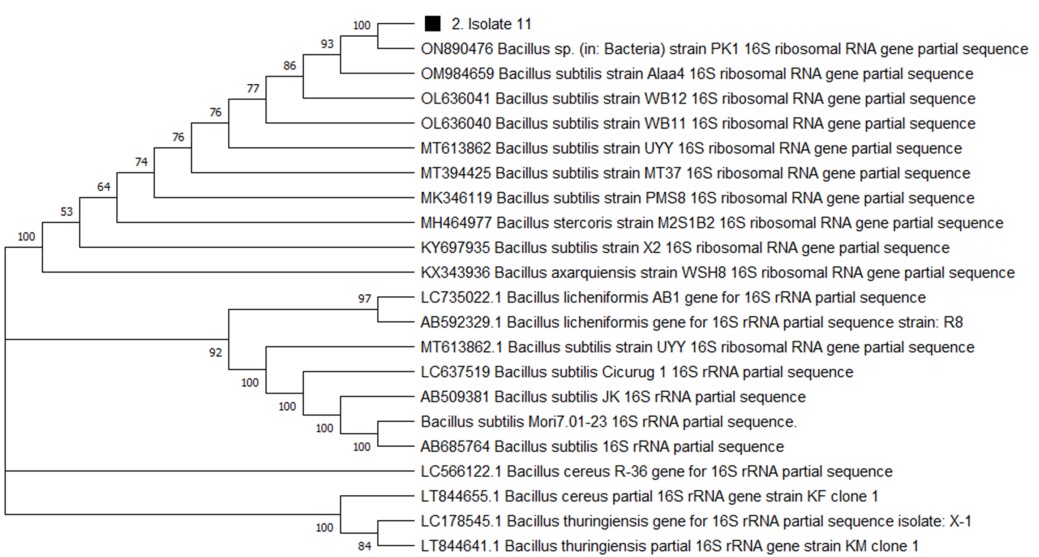

**Figure 3** **Phylogenetic tree based on the 16s RNA gene fragment.** Evolutionary relationship inferred using the Neighbour-Joining method. In this study, the soil isolate is represented by a squared black dot. The tree was built using a total of 21 DNA sequences taken at random from GenBank and labelled with their accession number. The analyses of evolution were conducted using MEGA X (10.0.5) (*Tamura, Stecher & Kumar, 2021*).

The metabolome captured in this study included both metabolites that were unknown and classes that were known or putatively annotated. With the aid of various databases, the data analysis resulted in the annotation of several compounds, these include Lipids, organic acid, alkaloids and derivatives, organoheterocyclic, benzenoids, and polyketides (Fig. S2). As detailed in the methodology, additional confirmatory scrutiny of the annotations was applied to ensure accuracy and reliability. This included NAP, MS2LDA and manually validated through MS-based accurate mass. Figure 5 shows the metabolites in their different clusters based on their structural similarities. The surfactin molecular family distinguished itself for exhibiting ions found only in the *Bacillus subtilis* species. In this cluster surfactins were annotated and nodes representing the ions at m/z 526.813, 540.829 and 1,072.49 (Fig. 5) were not annotated, indicating similarity of spectra between the aforementioned spectral nodes and the surfactins family. Furthermore, mass spectral datasets from all three bacterial strains, in silico annotation tools like dereplicator, MS2LDA and the network annotation propagation (NAP) were used to further investigate the chemical spaces of the various isolates.

## DISCUSSION

With the increasing number of antibiotic-resistant pathogens, it is becoming increasingly important to discover new antibacterial agents that are both highly active and have novel and diverse mechanisms of action. However, studies have been conducted from different sources in nature, for instance marine, terrestrial organisms and medicinal plants to combat the tide of antibiotic resistance (*Hayashi, Bizerra & Da Silva Junior, 2013*; *Doss et al., 2017*; *Watts*

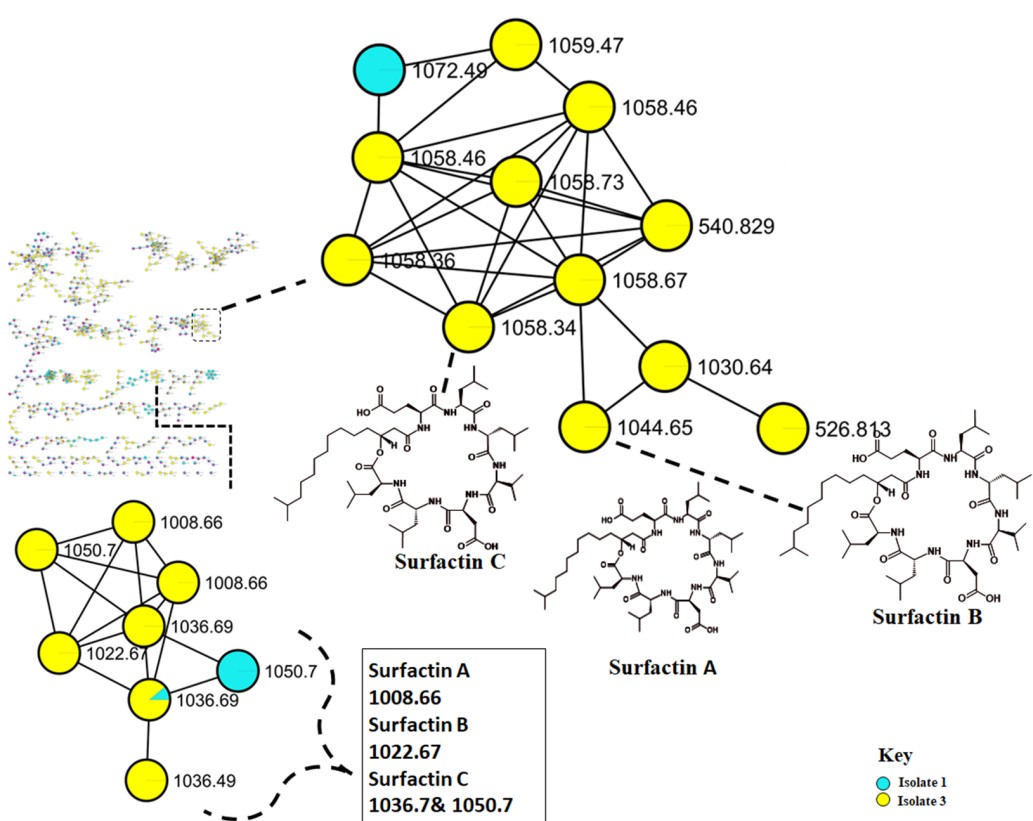

**Figure 4  Molecular network of *Bacillus* species extracts analysed through liquid chromatography-tandem mass spectrometry using electrospray ionisation (ESI) in positive mode.** Surfactins (Lipopeptides) clustering in different molecular families. Node colours represent the studied *Bacillus* species and the nodes in the network represent distinct molecular ions detected in the samples, while the edge connecting the node indicate spectral similarity between the compounds.

*et al., 2017*). Moreover, soil is a diverse and heterogeneous environment, characterized by significant variations in its physical, chemical, and biological properties (*Chandra & Kumar, 2017*). This poses challenges to microorganisms inhabiting the soil, prompting them to develop strategies for survival, such as production of antimicrobials. These antimicrobials can inhibit the growth of other microorganisms, providing a competitive advantage to the producing strain (*Sitotaw et al., 2022*). As such, the adaptation strategies they develop in response to this pressure make soil an ideal source for the isolation of antibiotic-producing bacteria. Additionally, natural products from microbial origin are becoming increasingly important over the last few decades, as they are more renewable, and maybe reproducible source than plants or animals (*Lam, 2007*). Hence, the current study aimed at isolation and characterization of microorganisms with the potential to produce antimicrobial compounds from soil from various agricultural settings.

The soil bacteria isolates obtained and evaluated for antimicrobial production against known pathogenic organisms were shown to exhibit substantial activity against the pathogenic test organism in the current study. As shown in Fig. 1, these bacterial isolates

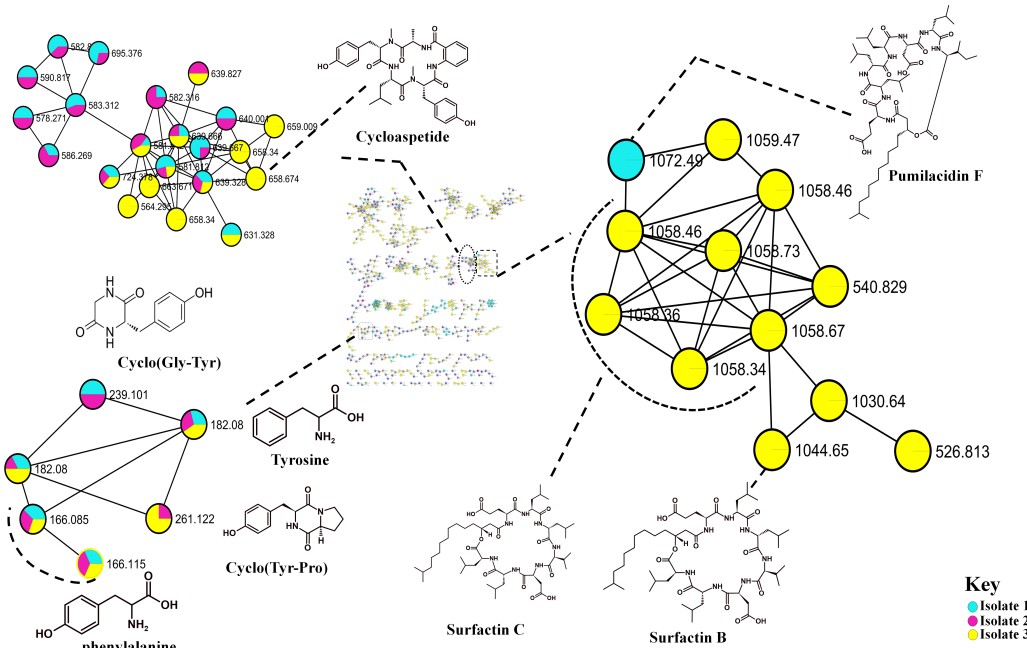

**Figure 5** Molecular network of *Bacillus* species extracts analysed through liquid chromatography-tandem mass spectrometry using electrospray ionisation in positive mode. Lipopeptides (right) and the cyclodipeptides (CDP) molecular family (left). Node colours represent the studied *Bacillus* species and the nodes in the network represent distinct molecular ions detected in the samples, while the edge connecting the node indicate spectral similarity between the compounds.

have demonstrated a broader spectrum of activities against various pathogens, with Isolate 3 (*Bacillus subtilis*) having the greatest inhibitory impact against *Staphylococcus aureus* ATCC (25923), whereas Isolate 1 had the least inhibitory effect against *Escherichia coli* ATCC (35218). This inhibition might be due to the presence of different secondary metabolites that have antimicrobial properties, the production of specific antimicrobial compounds, or differences in the mode of action. Nonetheless, our findings agree with the findings by *Sethi, Kumar & Gupta (2013)*, who also isolated *Bacillus subtilis* from soil which also showed to be active against some gram-positive including *Staphylococcus aureus* under standard culturing conditions.

The 16S rRNA sequencing plays a crucial role in advancing our understanding of microbial diversity, evolution, and the discovery of novel bacteria mostly in cases of unusual phenotypic profiles, uncultivable or rare bacteria (*Woo et al., 2008*). In the present study, 16S rRNA was performed to understand the evolutionary relationship and relatedness of different bacterial isolates. The identification of the three isolates was established by comparing them to the existing data in GenBank (NCBI) and cross-referencing with highly scored rRNA gene sequences in BLAST searches. The analysis of the amplified 16S rRNA gene through phylogenetics demonstrated that the two active isolates, henceforth referred to as Isolate 1 (3SIK) and Isolate 2 (8SIK), exhibit sequence homology with *Bacillus pumilus* (as illustrated in Fig. 2). In contrast, Isolate 3 displayed significant sequence homology with

*Bacillus subtilis* (as depicted in Fig. 3). These findings align with numerous prior studies indicating that *Bacillus* species are widespread and renowned for their antimicrobial production (*Khalid et al., 2016*).

Molecular networking analyses of UPLC-qTOF-MS data were conducted to unearth the chemistry diversity and classification of the metabolites produced by these bacterial isolates. Notably, these isolates produced a higher quantity of extracellular metabolites compared to intracellular metabolites (see Fig. S1). Moreover, to further investigate the chemical diversity of the bacterial isolates, the spectral data were analyzed through a publicly available database *via* the classical molecular networking workflow on the GNPS platform, enabling a broad overview of the chemical information deduced from the MS/MS data and the generated molecular networks were visualized in Cytoscape (Fig. S5) (*Doncheva et al., 2018*). As a result, this information was employed to further investigate into the identities of these nodes generated.

Molecular networking not only helps in annotating known compounds but also facilitates the discovery of novel metabolites. As such, the annotation of surfactins in the present study further facilitated with the identification or annotation of other nodes which were previously unidentified features at m/z 1072.49 which was in the same cluster with surfactin, and it was putatively identified as pumilacidin F (*Naruse et al., 1990*). This lipopeptide consists of a cyclic ring connected to a fatty acid chain and has been discovered to possess strong antimicrobial characteristics (*Jemil et al., 2017*). Thus far, several studies have demonstrated the antibacterial (*Saggese et al., 2018*), antifungal (*De Oliveira et al., 2020*), antiviral (*Singh & Cameotra, 2004*), antimotility activities (*Xiu et al., 2017*) of pumilacidin. *Saggese et al. (2018)* also showed that Pumilacidin has antimicrobial activities against *Staphylococcus aureus* and *Listeria monocytogenes*. Similarly, in the present study, isolate 1 showed to have convincing activity against *S. aureus*. Notably, in the cluster pumilacidin F was shown to be unique to *Bacillus pumilus*, which therefore gives confidence in the annotation of the compound (Fig. 5). Moreover, the findings from the present study suggest that the unannotated nodes within these clusters are structurally related to Surfactins, which can aid in the assignment of names to additional unannotated nodes. In addition, a combination of molecular network and manual confirmatory scrutiny, led us to identify another cyclodipeptide known as Isarrin D at m/z 540 $[M + H]^+$, a cyclodipeptides from the isaridine family (Fig. S4). It showed a fragment ion at m/z 427, m/z 330, m/z 211 and m/z 183. Isariins derivatives have been shown to have insecticidal (*Langenfeld et al., 2011*), antiplasmodial (*Lemmens-Gruber, Kamyar & Dornetshuber, 2009*) and anti-inflammatory activities (*Wang et al., 2018*).

The outcomes of high-throughput molecular networking did not match the extensive GNPS repository dataset in our study, giving an indication of the occurrence of new class of compounds among active isolates. It was also evident that not all bioactive metabolites (Lipopeptides) were clustering in the same cluster (Fig. 4), suggesting that there are some significant differences in the MS/MS fragmentation which therefore suggest the presence of different type of metabolites in the extracts. These isolates produce a diverse range of unidentified compounds which could be vital for the antimicrobial activities observed against test organisms (Fig. 1). This does, however, draw attention to one of the most

important drawbacks of untargeted metabolomics, as many metabolites are still unknown or poorly characterized particularly in microbial metabolomics (*Vinaixa et al., 2016*). These unknown metabolites are known as *dark matter* in metabolomics and efforts to generate algorithms such as those aimed at establishing the identity of these metabolites and the identity of the microbes that makes them are being made (*Wang et al., 2020*).

The recent development in the GNPS ecosystems has transformed the discovery of metabolites in high-throughput technology. The contribution of this *in silico* tools has enabled us to identify several cyclodipeptides such as Cyclo (Tyr-pro), Cyclo (Leu-pro), Cyclo (Pro-Val) and Cyclo (Pro-Phe) (Fig. S3). These cyclodipeptides compounds have relevance in quorum sensing (*González et al., 2017*), cell-cell signaling or as drug delivery systems (*Bojarska & Wolf, 2020*). Although, it is difficult to differentiate whether the observed response is due to cytotoxicity or bioactivity (*Liu et al., 2023*). The putative presence of these known bioactive compounds suggests that some of the antimicrobial activity might be attributed to these compounds. Overall, we were able to putatively identify several metabolites (Table S1) from *Bacillus* isolates using molecular networking approach. However, Isolate 3 had the greatest level of activity out of the three isolates, thus it can therefore be suggested that based on the results observed from the molecular network, the combination of the cyclopeptides and lipopeptides (Surfactins) together can confer an enhanced antimicrobial activity, and given that Isolates 3 and 8 produce these cyclodipeptide molecules instead of the lipopeptides. The other bacteria can withstand the effects of cyclodipeptides, as compared against the most potent surfactins. Therefore, these are resourceful and easy-to-find bacteria that have the potential to serve as a valuable source of antimicrobial compounds. Consequently, future bioreactors can be designed to cultivate these bacteria to produce high concentrations of the antimicrobial compounds. Moreover, stress inducer can be used to elevate the levels of this compounds.

## CONCLUSIONS

This study demonstrated that the use of (computational metabolomics tools) molecular networking approach is a useful tool in the field of bio-guided bioprospection of microorganisms and has proved to be an effective and feasible towards unearthing potent antimicrobial compounds. Our results demonstrated that the approach could accelerates the discovery of bioactive compounds and isolation of strains which offer a promising avenue for discovering new clinical antimicrobials. Amongst all the isolated bacteria, Isolate 3 (*Bacillus subtilis*) was found to be more potent. Therefore, in future bioreactors designed to cultivate these bacteria with stress inducers can be used to elevate the levels of these compounds. However, molecular networks generated from the methanol extracts enabled annotation of various metabolites, the existence of abundance of novel compounds which couldn't be identified (dark matter) has been noted and is major drawbacks of this approach, which could be remedied through a multi-analytical platform approach. Nonetheless, the notable activity observed and the chemical diversity from these two *Bacillus* species, *Bacillus pumilus* and *Bacillus subtilis* motivate the search for new antimicrobial metabolites from these two species. Additionally, these could be a useful

starting point for the discovery of clinically significant and new antimicrobial agents in the future.

## ACKNOWLEDGEMENTS

The University of Venda, Department of Biochemistry and Microbiology are gratefully thanked for access to the LC-qTOF-MS. SHIMADZU-SA is thanked for the technical support. Ms. A-T Ramabulana is thanked for continuous discussion on molecular networking.

### Funding

The work was funded by the National Research Foundation of South Africa. The funders had no role in study design, data collection and analysis, decision to publish, or preparation of the manuscript.

### Grant Disclosures

The following grant information was disclosed by the authors:
National Research Foundation of South Africa.

### Competing Interests

The authors declare there are no competing interests.

### Author Contributions

- Adivhaho Khwathisi conceived and designed the experiments, performed the experiments, analyzed the data, prepared figures and/or tables, authored or reviewed drafts of the article, and approved the final draft.
- Ntakadzeni Edwin Madala conceived and designed the experiments, performed the experiments, analyzed the data, prepared figures and/or tables, authored or reviewed drafts of the article, and approved the final draft.
- Afsatou Ndama Traore analyzed the data, prepared figures and/or tables, authored or reviewed drafts of the article, and approved the final draft.
- Amidou Samie conceived and designed the experiments, analyzed the data, prepared figures and/or tables, authored or reviewed drafts of the article, and approved the final draft.

### Data Availability

The chromatographic raw data are available at:

- METABOLOMICS-SNETS-V2: https://gnps.ucsd.edu/ProteoSAFe/status.jsp?task=29efd5676d4f44fb82a8b1831011f43b.

- MS2LDA_MOTIFDB:

https://gnps.ucsd.edu/ProteoSAFe/status.jsp?task=5eb48bbacbfb44a2946b119041566590;

- MOLNETENHANCER: https://gnps.ucsd.edu/ProteoSAFe/status.jsp?task=97c433b7506d4b33a6099ba150ad3d37;
- NAP_CCMS2: https://gnps.ucsd.edu/ProteoSAFe/status.jsp?task=bee7bdc7eccf49c2a6d6a08ffffaf7be;
- DEREPLICATOR_PLUS: https://gnps.ucsd.edu/ProteoSAFe/status.jsp?task=2f8b61d505b74a678f205962e0a308ae

The raw chromatographic data are available at Zenodo: Khwathisi, A. (2024). Bioprospecting of soil-borne microorganisms and chemical dereplication of their anti-microbial constituents with the aid of UPLC-QTOF-MS and Molecular Networking approach [Data set]. In Bioprospecting of soil-borne microorganisms and chemical dereplication of their anti-microbial constituents with the aid of UPLC-QTOF-MS and Molecular Networking approach. Zenodo. https://doi.org/10.5281/zenodo.11068876.

## Supplemental Information

Supplemental information for this article can be found online at http://dx.doi.org/10.7717/peerj.17364#supplemental-information.

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
