# Peer review of "Bioprospecting of soil-borne microorganisms and chemical dereplication of their anti-microbial constituents with the aid of UPLC-QTOF-MS and molecular networking approach"

_PeerJ, doi:10.7717/peerj.17364_

## Round 0.1 · original submission · Minor Revisions

The manuscript was reviewed by two experts. Their comments are largely positive. Reviewer #1 raises several points, most of which are requests for clarification. Reviewer #2 requests that you include a positive control for the anti-bacterial assays, as well as further information on the isolated strains. Please address these comments in a revised version of your paper.

**Language Note:** The review process has identified that the English language must be improved. PeerJ can provide language editing services - please contact us at [email protected] for pricing (be sure to provide your manuscript number and title). Alternatively, you should make your own arrangements to improve the language quality and provide details in your response letter. – PeerJ Staff

Reviewer 1 ·

Basic reporting

The manuscript entitled "Bioprospecting of soil-borne microorganisms and chemical dereplication of their anti-microbial constituents with the aid of UPLC-QTOF-MS and Molecular Networking approach" is well written and reports significant results. The study involved the recent/modern screening techniques "molecular networking" to predict the presence of various bio active compounds in the fermentation broth of the isolated strains. The authors needs to incorporate the following revisions in to the manuscript before proceeding further

1. Lines 60-61. Change "Finally, soil could serve an alternative source of anti-microbial compounds" to "Finally, soil microbial flora could serve an alternative source of anti-microbial compounds"
2. Line 136: Correct "withing" as "within"
3. Line 148: Sentence needs correction and wording is ambiguous, for example "microbial compounds which may be for use in the management of" ??
4. Line 157: "(-23.083019, 30.301499)" add proper degree signs and directions on GPS data
5. Line 167: "Jamil et al., (2007)" check this reference, its not related to bacterial isolation
6. Line 203: "(dH2O)" use proper formula format, 2 as subscript
7. Lines 211-221: The quantity of broth used to cultivate the strains needs to be mentioned clearly, MH broth is rarely used for metabolites production?, it could be nutrient broth or LB broth for general bacteria, the extraction procedure for liquid phase and solid phase cell pellet is ambiguous, mainly the extraction of cell pellet is described, how the supernatant was extracted, because the direct addition of methanol to water will not separate the two solvents afterwards, clearly describe this procedure
8. Line 232: "The gradient technique was gradually increased from 3 to 30 minutes" change to "The solvent gradient was set from 3 to 30 minutes"
9. Lines 287-296: The values for prominent zone of inhibition should be mentioned in the text
10. Fig 2: The phylogenetic tree fig is invisible needs clear picture
11. Lines 264-280: The method explained here is important and although satisfactory, but missing a key point, where is the information of blank?, seems like compounds present in the blank (culture media/broth) have also been claimed to be produced by the bacterial isolates
12. Lines 303-304: Please mention the nanodrop value of extracted DNA, especially what was the exact concentration of DNA, and also mention the A260/280 values
13. Line 423: "Bacillus species" italicize the genus name
14. The bioactivity of the extracts of the strains could be due to the general cytotoxicity of the metabolites, however the bioassays data lack the cytotoxicity data of the bacterial extracts, what is author's stance on it, further in discussion the bioactivity and cytotoxicity of the extracts should be correlated with the compounds detected by molecular networking

Experimental design

Experimental design is good and support the outcome of the study

Validity of the findings

The number of samples analysed is bit small, however the data seems sufficient for a routine scientific paper. The results reported are significant and sound valid

Additional comments

1. In the figure legends and labels the scientific names of the organisms should be italics
2. The compounds structures in Figs 4 and 5 are invisible

Reviewer 2 ·

Basic reporting

Ther are some spelling mistakes; the manuscript should be checked thoroughly.

Experimental design

In the anti-bacteria test, is there any positive control used? And the photos of the plates should be provided in the Supplementary material.

Validity of the findings

In the result section, the description of strains isolated from soil should be provided, such as: the numbers of the isolates, the species of them. And the photos of the isolated strains were suggested to be provided in the Supplementary material.

Additional comments

no comment

---

## Round 0.2 · accepted · Accept

The revised manuscript has satisfactorily addressed the comments raised by the reviewers.

Reviewer 1 ·

Basic reporting

The authors have incorporated the required changes and responded to most the revisions.

Experimental design

OK

Validity of the findings

Ok

Reviewer 2 ·

Basic reporting

no comment

Experimental design

no comment

Validity of the findings

no comment